# Differentiating between Lymphoma and Metastasis Presenting as Solid Cerebellar Mass Lacking Necrosis

**DOI:** 10.3390/diagnostics14192228

**Published:** 2024-10-06

**Authors:** Gye Ryeong Park, Byung Hyun Baek, Seul Kee Kim, Woong Yoon, Ilwoo Park, Yun Young Lee, Tae-Young Jung

**Affiliations:** 1Department of Radiology, Chonnam National University Medical School, Chonnam National University Hospital, Gwangju 61469, Republic of Korea; pgr1025@gmail.com (G.R.P.); radyoon@jnu.ac.kr (W.Y.); ipark@jnu.ac.kr (I.P.); yunyoung0219@gmail.com (Y.Y.L.); 2Department of Radiology, Chonnam National University Medical School, Chonnam National University Hwasun Hospital, Hwasun 58128, Republic of Korea; kimsk.rad@gmail.com; 3Department of Neurosurgery, Chonnam National University Medical School, Chonnam National University Hwasun Hospital, Hwasun 58128, Republic of Korea; jung-ty@jnu.ac.kr

**Keywords:** lymphoma, metastasis, cerebellum, magnetic resonance imaging

## Abstract

**Objectives:** This study aimed to identify radiologic features that differentiate lymphoma from metastasis manifesting as a solid enhancing mass lacking necrosis in the cerebellum. **Methods:** Pathologically confirmed 24 primary central nervous system lymphoma (PCNSL) and 32 metastasis patients with solid enhancing cerebellar masses without necrotic or hemorrhagic components were retrospectively analyzed. We evaluated the imaging characteristics using contrast-enhanced magnetic resonance imaging (MRI). The serrate sign was defined as a tumor spreading along white matter with branch-like enhancement or outward spikes. **Results:** The serrate sign was exclusively identified in the PCNSL group, showing a significant difference compared to the metastasis group (75.0% vs. 0%, *p* < 0.001). Homogeneous enhancement occurred more frequently in PCNSL than in metastasis (91.7% vs. 21.9%, *p* < 0.001). Conversely, bulging contour (62.5% vs. 4.2%, *p* < 0.001) and surface involvement (71.9% vs. 29.2%, *p* = 0.003) were more prevalent in metastasis than PCNSL. For predicting PCNSL, the serrate sign demonstrated 75.0% sensitivity, 100% specificity, 100% positive predictive value, 84.2% negative predictive value, and 89.3% accuracy. **Conclusions:** This study found that the serrate sign and homogeneous enhancement are reliable MRI features for differentiating cerebellar PCNSL from metastasis, whereas a bulging contour and surface involvement suggest metastasis. The serrate sign demonstrated diagnostic significance in differentiating PCNSL from metastasis.

## 1. Introduction

Primary central nervous system lymphoma (PCNSL), a rare subtype of extranodal non-Hodgkin lymphoma, accounts for 1–3% of all central nervous system (CNS) tumors. PCNSL located in the cerebellum is even less common, occurring in only 5.1–9.4% of cases [1,2]. The overall incidence rate of PCNSL is approximately 4.7 per 1,000,000 person-years, with a slight increase in the elderly population post-2000. Despite this, survival rates remain poor [3]. Histopathological confirmation is crucial for PCNSL diagnosis before treatment initiation. The gold standard for diagnosis is stereotactic needle biopsy, ideally conducted without prior steroid use [4]. Differentiating PCNSL from other central nervous system tumors is vital, typically using preoperative brain magnetic resonance imaging (MRI).

Brain metastatic tumors are the most common CNS tumors in adults, including those in the cerebellum. Cerebellar metastatic tumors often necessitate aggressive treatments such as surgical resection, radiosurgery, or steroid therapy, especially due to the risks of obstructive hydrocephalus and brain stem compression, which can be life-threatening [5,6,7]. The infrequency of necrosis in PCNSL among immunocompetent patients poses challenges in distinguishing it from metastatic tumors [1,2], particularly when a cerebellar tumor appears as a solid enhancing mass without necrosis or hemorrhagic components. This differentiation is critical due to the significantly different treatment approaches required for PCNSL and cerebellar metastatic tumors.

Several studies have documented the brain imaging characteristics of PCNSL in immunocompetent patients [1,2,8,9,10,11,12]. However, detailed insights into the specific MRI features of PCNSL affecting the cerebellum are scarce. Notably, a branch-like enhancement pattern has recently been identified as a distinct feature of cerebellar lymphoma, differentiating it from high-grade glioma and other pathologies [12,13]. Previous studies included cerebellar masses with cystic and hemorrhagic components. Rare cases of metastatic tumors that do not contain necrosis or hemorrhage can present a diagnostic challenge, particularly in the cerebellum, where they can be difficult to distinguish from PCNSL. The differentiation of cerebellar tumors, specifically focusing on non-necrotic solid masses, as seen in PCNSL and rare types of metastatic tumors, has not been studied.

The cerebellum is uniquely structured with complex, intricate inward folding of its cortex, known as foliation [14]. We theorized that the dense foliation of the cerebellum, combined with the infiltrative nature of PCNSL, might influence the tumor’s morphological characteristics in this region. Specifically, we proposed that PCNSL situated in the central cerebellar white matter, which is distant from the foliation, would be less likely to show branch-like enhancement. Instead, central white matter PCNSL might invade only the root of the foliation area, resulting in an outward spike pattern. Consequently, we introduced the term “serrate sign” to describe lesions exhibiting branch-like enhancement or outward spike patterns following white matter structures. This study aimed to retrospectively evaluate preoperative contrast-enhanced MRI to explore the diagnostic value of the serrate sign and other radiological features for distinguishing PCNSL from metastasis in patients with solid enhancing cerebellar masses.

## 2. Materials and Methods

### 2.1. Subjects

From May 2004 to December 2022, we conducted a retrospective review of medical records and preoperative MR imaging for patients with pathologically confirmed PCNSL and brain metastasis. These patients had undergone brain surgery or biopsy. Figure 1 presents a flowchart of patient inclusion and exclusion. Of the 194 patients with PCNSL, 39 had cerebellar lesions. We excluded those with specific cerebellar lesions: subependymal involvement (*n* = 5), angiocentric pattern (*n* = 1), brain stem lesion extending to the middle cerebellar peduncle (*n* = 2), and lesions smaller than 1 cm (*n* = 7). In the group of 591 patients with brain metastasis, 138 had cerebellar masses. From this group, we excluded patients with cerebellar metastatic lesions containing necrotic/hemorrhagic components (*n* = 86) and lesions smaller than 1 cm (*n* = 20). After reviewing MR images, we enrolled 24 patients with PCNSL and 32 with metastasis in the study. We collected clinical data from these 56 patients, including age, sex, immune status, underlying malignancy at the time of surgery, surgical method for specimen sampling, and history of lymphoma at least 6 months before surgery. The institutional ethics committee approved the study and waived the requirement for informed consent due to its retrospective nature.

### 2.2. Brain MR Imaging Analysis

All patients underwent preoperative brain MRI examinations using 1.5 T and 3.0 T systems. Following an intravenous injection of gadolinium-based contrast, contrast-enhanced T1WI was acquired in three orthogonal planes, and the slice thickness of each was noted. Two patients received images in only two orthogonal planes, comprising axial with coronal and axial with sagittal images. Conventional MRI scans were evaluated retrospectively by two neuroradiologists, who were blinded to the pathological findings. Disagreements were resolved through consensus. The analysis focused on the largest cerebellar lesion in each patient, using contrast-enhanced T1WI images. Detailed imaging findings included the following: (1) size (maximum diameter in millimeters), (2) lesion location (hemisphere, vermis, or cerebellar peduncle), (3) distribution (number of tumors and presence of concomitant supratentorial lesion), (4) enhancement pattern (homogeneous or heterogeneous), (5) morphological features including the presence of the serrate sign, defined as the tumor displaying a branch-like enhancement pattern or outward spikes (Figure 2), observable on any orthogonal plane, (6) bulging contour of the tumor (Figure 3), and (7) cerebellar surface involvement by the tumor. Axial T2WI or fluid-attenuated inversion recovery images were reviewed to evaluate peritumoral streak-like edema. The study investigated associations between tumor size and morphologic features among patients with PCNSL and metastasis.

### 2.3. Statistical Analysis

We compared demographic and MRI characteristics between the PCNSL group and the metastasis group. The chi-square test or Fisher’s exact test was used for categorical variables, and the Mann–Whitney U test was used for continuous variables. The analysis was conducted using IBM SPSS Statistics for Windows (version 27.0; IBM Corp., Armonk, NY, USA). Variables with *p*-values less than 0.05 were considered significant. The diagnostic value of the serrate sign in predicting PCNSL, including sensitivity, specificity, positive predictive value, negative predictive value, and accuracy, was calculated.

## 3. Results

This study included 56 patients (33 men and 23 women) with solid enhancing cerebellar tumors. Clinical and demographic characteristics are detailed in Table 1. No significant age or gender differences were noted between the PCNSL group and the metastasis group. All participants were immunocompetent. Within the PCNSL group, two patients had a history of gastrointestinal malignancies and one of prostate cancer. Additionally, four patients had a history of remission from previously diagnosed lymphoma. PCNSL diagnosis was confirmed as diffuse large B-cell lymphoma in all cases, either through stereotactic biopsy (83.3%) or partial tumor removal (16.7%), involving 16 cerebellar and 8 supratentorial lesions. The metastasis group included diverse primary malignancies as follows: lung cancer (14 patients), gastric cancer (5), colorectal cancer (3), breast cancer (8), prostate cancer (1), and hepatocellular carcinoma (1). None in the metastasis group had a history of lymphoma. All metastatic cases were diagnosed via surgical resection.

Table 2 presents the comparison of MRI findings between the two groups. No significant differences were found in tumor size, location, or distribution. Homogeneous enhancement was more common in the PCNSL group (91.7%) than in the metastasis group (21.9%, *p* < 0.001). The serrate sign appeared in 18 of 24 PCNSL patients but was absent in the metastasis group (75% vs. 0%, *p* < 0.001). Of the 18 PCNSL patients with the serrate sign, 13 exhibited a branch-like enhancement pattern, and 5 showed outward spikes without branch-like enhancement. Among these five cases, four were located in the central white matter or middle cerebellar peduncle. Conversely, a bulging contour was more frequent in metastasis than in PCNSL (62.5% vs. 4.2%, *p* < 0.001). Cerebellar surface involvement was also more common in metastasis (71.9% vs. 29.2%, *p* = 0.003). No significant difference was observed in perilesional streak-like edema between the groups.

Detailed patient characteristics of the PCNSL and metastasis groups are provided in Appendix A, respectively. In the PCNSL group, the serrate sign was observed more on sagittal (70.8%, 17/24) than coronal (45.8%, 11/24) and axial (37.5%, 9/24) images (Appendix A). Associations of image features and tumor size in cerebellar PCNSL and metastasis patients are presented in Table 3. Tumors with the serrate sign were significantly larger than those without it in the PCNSL group (mean, 2.94 cm vs. 1.95 cm, *p* = 0.009). In the metastasis group, tumors with a bulging contour were larger than those without it (mean, 3.79 cm vs. 2.69 cm, *p* = 0.007). Similarly, tumors with cerebellar surface involvement were larger than those without it in both groups: in PCNSL (mean, 3.41 cm vs. 2.40 cm, *p* = 0.02) and in metastasis (mean, 3.59 cm vs. 2.8 cm, *p* = 0.04). Conversely, tumors with homogeneous enhancement were significantly smaller than those with heterogenous enhancement in the PCNSL group (mean, 2.94 cm vs. 4.3 cm, *p* = 0.036). For predicting PCNSL, the serrate sign demonstrated a sensitivity of 75.0%, a specificity of 100%, a positive predictive value of 100%, a negative predictive value of 84.2%, and an accuracy of 89.3%.

## 4. Discussion

The present study revealed that the serrate sign is a highly specific indicator with a significant positive predictive value for diagnosing cerebellar PCNSL, distinguishing it from non-necrotic solid enhancing metastases. Additionally, we observed that homogeneous enhancement is more commonly associated with PCNSL than with metastasis, while bulging contour and cerebellar surface involvement are more frequently seen in metastasis than in PCNSL.

The serrate sign was exclusively identified in the PCNSL group, occurring in 75% of cases. This sign has been proposed as a revised term for branch-like enhancement, reflecting the phenomenon where PCNSL located in the central white matter may exhibit outward spikes. In our study, 5 out of 18 patients exhibiting the serrate sign showed outward spikes without branch-like enhancement. Most PCNSL lesions with outward spikes were found in central white matter structures. He et al. recently reported that a branch-like enhancement pattern in cerebellar tumors is a useful imaging feature for differentiating PCNSL from high-grade glioma, occurring along white matter fibers in 66.7% of PCNSL cases [12]. Another study by Yokoyama et al. showed that branch-like enhancement is a specific indicator of cerebellar lymphoma compared to other pathologies, observed in 88% of cerebellar lymphoma cases [13]. In contrast to our study, which focused on non-necrotic solid enhancing masses, their study’s control group mainly comprised metastases, many of which had cystic or hemorrhagic components. The findings of our research, alongside previous studies, suggest that the serrate sign results from the infiltrative nature of PCNSL along white matter fibers. Previous studies have reported that the most common location for PCNSL in the central nervous system is the cerebral white matter and corpus callosum, leading to distinctive imaging features such as butterfly and notch signs in the supratentorial region [1,2,9,10]. From an anatomical perspective, the unique foliation of the cerebellum is presumed to contribute to the formation of the serrate sign, and the occurrence of outward spikes in central white matter PCNSL can be attributed to their location away from this foliation.

In our study, the serrate sign appeared more frequently on sagittal images (70.8%, 17/24) than on coronal (45.8%, 11/24) and axial images (37.5%, 9/24). This finding aligns with Yokoyama et al.’s observation that branch-like enhancement predominantly occurs in sagittal images in three-dimensional contrast-enhanced T1WI [13]. However, the reasons behind this phenomenon have not been thoroughly explored. Considering cerebellar anatomy, the higher detection rate of the serrate sign in sagittal images may be due to the predominant foliation, or folding, in the rostrocaudal direction, which is a result of the elongated cerebellar cortex in this orientation [15]. Additionally, the thinner section thickness in sagittal images (mean: 2.89 mm) compared to that in axial and coronal images (mean: 4.78 mm and 4.79 mm, respectively) in our study likely contributed to higher visibility rates. Consequently, our findings indicate that a detailed examination of sagittal plane images in contrast-enhanced T1WI is crucial for effectively characterizing cerebellar PCNSL, emphasizing the importance of acquiring images with thin section thickness.

An infiltrative growth pattern is a recognized imaging characteristic of PCNSL. The infrequent surface involvement of PCNSL observed in our study is presumably linked to the tendency of PCNSL to spread along white matter tracts. Nonetheless, seven patients with PCNSL exhibited cerebellar surface involvement. In the PCNSL group, tumors with surface involvement were significantly larger than those without it. Therefore, even in PCNSL cases, larger masses may invade the cerebellar surface.

Homogeneous enhancement on contrast-enhanced imaging is a common characteristic of PCNSL in immunocompetent patients, whereas in immunocompromised patients, PCNSL often presents with a heterogeneous enhancement pattern, frequently featuring ring enhancement [8,9,10,11,16]. In our study, all participants were immunocompetent at the time of presentation, and homogeneous enhancement was observed more often in the PCNSL group compared to the metastasis group (91.7% vs. 21.9%, *p* < 0.001). In PCNSL, patients with heterogeneous enhancement had larger tumors compared to those with homogeneous enhancement. Even in cases of PCNSL, it is important to consider that larger tumors may exhibit heterogeneous enhancement. Streak-like edema has been reported as a useful marker for distinguishing cerebellar PCNSL from high-grade glioma [12]. In our research, streak-like edema was prevalent in both PCNSL and metastasis cases. In line with our findings, Yokoyama et al. reported that most patients with cerebellar lymphoma and metastasis exhibited streak-like edema [13].

Interestingly, we noted a bulging contour of the tumor, a feature we identified as contrasting with the serrate sign indicative of an infiltrative nature. This feature was significantly more common in metastasis than in PCNSL. This difference likely stems from the distinct origins of these conditions. PCNSL originates from lymphoid tissue within the white matter, whereas metastasis involves the invasion of external cells. As a metastatic lesion expands, it tends to produce the bulging tumor contour observed in our study. In the metastasis group, tumors with a bulging contour were larger in size compared to those without it.

Accurate preoperative diagnosis is crucial for suspected PCNSL patients, as steroids should be avoided prior to biopsy to prevent interference with histopathologic diagnosis. In contrast, cerebellar metastatic tumors often require surgical intervention, especially for decompression, while surgical resection typically offers no substantial advantage over chemotherapy in treating PCNSL, except in cases of rapid neurological decline or impending brain herniation [17,18]. Distinguishing between PCNSL and metastasis preoperatively poses a challenge, particularly when a solid enhancing mass is present in the cerebellum without necrotic or hemorrhagic components. Our study is the first to investigate radiologic features that differentiate PCNSL from metastasis, the most common type of cerebellar tumor, in patients with non-necrotic solid enhancing cerebellar masses using contrast-enhanced MRI.

The study presented several limitations, including a small sample size and a retrospective design. In particular, caution is needed when interpreting the association between imaging features and tumor size, as the sample size in each individual group was small. Therefore, further studies with larger sample sizes are needed to validate these findings. Additionally, not all cerebellar lesions were assessed, as lesions smaller than 1 cm were challenging to characterize, particularly in cases of PCNSL. Despite this, our study included 24 cases of cerebellar PCNSL, representing, to our knowledge, the largest reported cohort with imaging features. Another significant limitation was the heterogeneity of MRI acquisition parameters, with some datasets comprising images captured using outdated MR equipment and transferred scans from external hospital MR scanners. Notably, advanced MR imaging techniques like diffusion-weighted imaging, perfusion-weighted imaging, and diffusion tensor imaging, known to aid in differentiating PCNSL from other tumors, were not evaluated in this study because only a limited number of patients, particularly in the metastasis group, underwent these advanced imaging techniques [19,20,21]. In PCNSL, patients exhibiting the serrate sign had larger tumors compared to those without this feature. Therefore, extreme caution is advised when evaluating smaller-sized cerebellar lesions.

## 5. Conclusions

Our findings indicate that the presence of the serrate sign and homogeneous enhancement are valuable MR characteristics for distinguishing cerebellar PCNSL from solid enhancing metastasis. Conversely, bulging contour and surface involvement were more suggestive of metastasis than PCNSL. The serrate sign demonstrated diagnostic significance in differentiating PCNSL from metastasis. Utilizing thin section thickness and focusing on the sagittal plane in contrast-enhanced MRI could enhance the identification of the serrate sign.

## Figures and Tables

**Figure 1 diagnostics-14-02228-f001:**
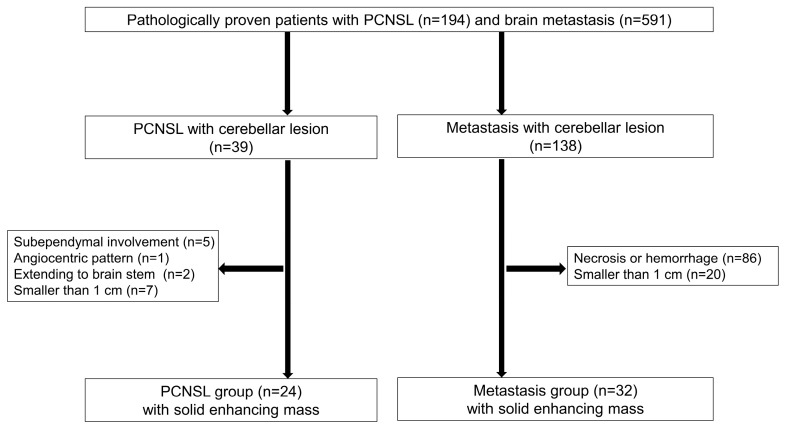
Flow chart detailing the inclusion and exclusion of patients.

**Figure 2 diagnostics-14-02228-f002:**
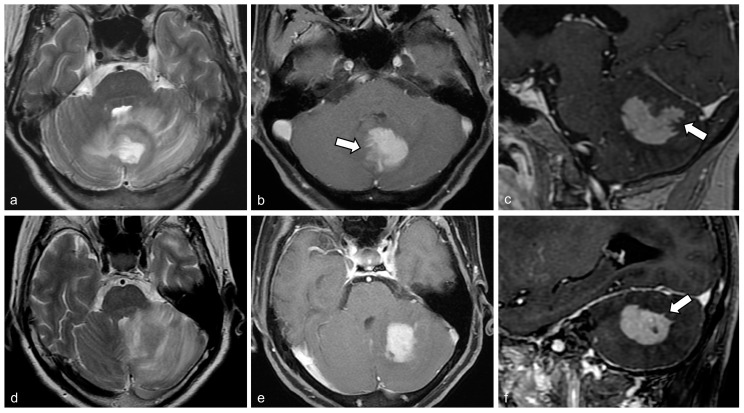
The typical “serrate sign” associated with primary central nervous system lymphoma (PCNSL) in the cerebellum. Panels a-c depict images from a 60-year-old female patient with cerebellar PCNSL. An axial T2-weighted image (**a**) displays peritumoral streak edema. Contrast-enhanced T1-weighted axial (**b**) and sagittal (**c**) images reveal a homogeneously enhancing mass in the central and peripheral white matter involving foliation, characterized by the serrate sign, which exhibits a branch-like enhancement pattern (arrows). Panels (**d**–**f**) show images from a 74-year-old female patient with cerebellar PCNSL. An axial T2-weighted image (**d**) also displays peritumoral streak edema. An axial contrast-enhanced T1-weighted image (**e**) shows a homogeneously enhancing mass involving central white matter, while a sagittal image (**f**) reveals the serrate sign featuring outward spikes (arrow).

**Figure 3 diagnostics-14-02228-f003:**
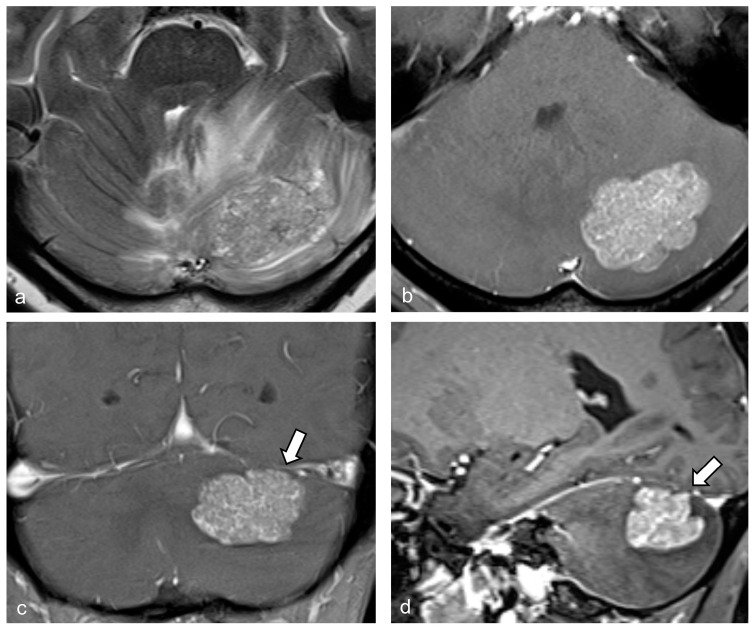
Images from a 55-year-old female patient with brain metastasis of breast cancer. An axial T2-weighted image (**a**) shows peritumoral streak edema in both cerebellar hemispheres. Contrast-enhanced T1-weighted axial (**b**), coronal (**c**), and sagittal (**d**) images display a non-necrotic, heterogeneously enhancing mass with a bulging contour and surface involvement (arrows).

**Table 1 diagnostics-14-02228-t001:** Clinical and demographic characteristics of cerebellar PCNSL and metastasis patients.

Patient Data	PCNSL (*n* = 24)	Metastasis (*n* = 32)
Age (mean ± SD), y	64.0 ± 10.2	58.2 ± 12.6
Sex, male	17 (70.8)	16 (50.0)
Immune status		
Immunocompetent	24 (100.0)	32 (100.0)
Immunocompromised	0 (0)	0 (0)
Underlying malignancy		
Lung malignancy		10 (adenocarcinoma) 4 (others)
Gastric malignancy	1 (adenocarcinoma)	5 (adenocarcinoma)
Colorectal malignancy	1 (adenocarcinoma)	3 (adenocarcinoma)
Breast malignancy		7 (invasive ductal carcinoma) 1 (mucinous carcinoma)
Others	1	2
Previous remission history of lymphoma	4 (16.6)	0 (0.0)
Pathologic procedure		
Biopsy	20 (83.3)	0 (0.0)
Resection	4 (16.7)	32 (100.0)

PCNSL, primary central nervous system lymphoma.

**Table 2 diagnostics-14-02228-t002:** Comparison of MR findings for cerebellar PCNSL and metastasis patients.

Patient Data	PCNSL (*n* = 24)	Metastasis (*n* = 32)	*p* Value
Size (mean ± SD) mm	27.0 ± 0.93	33.7 ± 1.13	0.22
Location			1.00
Hemisphere	21 (87.5)	27 (84.4)	
Vermis	3 (12.5)	6 (18.8)	
Middle cerebellar peduncle	2 (8.3)	0 (0)	
Distribution of tumors			
Single lesion	8 (33.3)	12 (37.5)	0.79
Concomitant supratentorial lesions	10 (41.7)	18 (56.3)	0.42
Homogeneous enhancement	22 (91.7)	7 (21.9)	<0.001
Serrate sign	18 (75.0)	0 (0)	<0.001
Branch-like enhancement	13 (54.2)		
Outward spikes	5 (20.8)		
Bulging contour	1 (4.2)	20 (62.5)	<0.001
Surface involvement	7 (29.2)	23 (71.9)	0.003
Streak-like edema	20 (83.3)	29 (90.6)	0.45

PCNSL, primary central nervous system lymphoma.

**Table 3 diagnostics-14-02228-t003:** Association of image features and tumor size in cerebellar PCNSL and metastasis patients.

	PCNSL (*n* = 24)		*p* Value	Metastasis (*n* = 32)		*p* Value
	Serrate sign (+), *n* = 18	Serrate sign (−), *n* = 6		Serrate sign (+), *n* = 0	Serrate sign (−), *n* = 32	
Size (mean ± SD) cm	2.94 ± 0.90	1.95 ± 0.63	0.009	0	3.36 ± 1.13	NA
	Homogeneous enhancement (+), *n* = 22	Homogeneous enhancement (−), *n* = 2		Homogeneous enhancement (+), *n* = 7	Homogeneous enhancement (−), *n* = 25	
Size (mean ± SD) cm	2.94 ± 0.83	4.3 ± 0.28	0.036	2.77 ± 1.00	3.53 ± 1.11	0.121
	Bulging contour (+), *n* = 1	Bulging contour (−), *n* = 23		Bulging contour (+), *n* = 20	Bulging contour (−), *n* = 12	
Size (mean ± SD) cm	3.4	2.67 ± 0.94	0.347	3.79 ± 1.18	2.69 ± 2.58	0.007
	Surface involvement (+), *n* = 7	Surface involvement (−), *n* = 17		Surface involvement (+), *n* = 23	Surface involvement (−), *n* = 9	
Size (mean ± SD) cm	3.41 ± 0.90	2.4 ± 0.79	0.024	3.59 ± 1.17	2.8 ± 0.85	0.04

PCNSL, primary central nervous system lymphoma; NA, not available.

## Data Availability

The data that support the findings of this study are available from the corresponding author upon reasonable request.

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
