# Peer review of "Differentiating between Lymphoma and Metastasis Presenting as Solid Cerebellar Mass Lacking Necrosis"

_diagnostics, 2024, doi:10.3390/diagnostics14192228_

Round 1

Reviewer 1 Report

Comments and Suggestions for Authors

Dear Authors, 

Thank you for your valuable work.

I am not sure about the statistical analysis of the Table#3. I think for some subjects, the number of patients are too small to analyze.It should be re-evaluated by an analyst or explained in a paragraph.

Kind Regards,

Comments on the Quality of English Language

Minor English editing required.

Author Response

Comment 1: I am not sure about the statistical analysis of the Table#3. I think for some subjects, the number of patients are too small to analyze.It should be re-evaluated by an analyst or explained in a paragraph.

Response 1: We agree with your point that caution is needed when interpreting the results of the analysis in Table 3. To investigate the association between tumor size and the signs we presented, we performed statistical analysis using the Mann-Whitney U test, which is useful when the data do not follow a normal distribution or when sample sizes are small. We re-checked the results using the Mann-Whitney U test and confirmed the same outcomes. As you pointed out, we acknowledge the limitations posed by the small sample sizes when analyzing the presence or absence of signs within the PCNSL and metastasis groups. (For example, only two patients with PCNSL showed non-homogenous enhancement) However, we believe that the clinical significance of the signs in relation to tumor size, as discussed in the manuscript, remains valid along with the statistical significance. We have also added a note in the limitations section regarding the potential uncertainty due to the small sample sizes, as you suggested.

- Changes in manuscript (disscusion, line262-264)

“The study presented several limitations, including a small sample size and a retrospective design. In particular, caution is needed when interpreting the association between imaging features and tumor size, as the sample size in each individual group was small. Therefore, further studies with larger sample sizes are needed to validate these findings.“

Response to minor editing of English language required: We have corrected the spacing issues in Table 3 and addressed the errors in quotation marks in the Conclusion section.

Reviewer 2 Report

Comments and Suggestions for Authors

I applaud the authors in clearly presenting a possible new radiologic biomarker to improve pretest probability in making an accurate diagnosis of PCNSL. The serrate sign here is one that ought be easily appreciated by trained neuroradiologists, neurologists, and neurosurgeons. Identifying such imaging patterns that can be easily taught to trainees and incorporated in daily practice with routine brain MRIs. 

The authors mention in line 239-240 that homogenous enhancement is characteristic of PCNSL in immunocompetent patients, however, directly stating that immunocompromised PCNSL are more likely to have ring enhancement may add slightly to the educational aspect of this manuscript. It is not a necessary addition to make but the difference is worthwhile for clinicians to be made aware of. 

Author Response

Comment 1: I applaud the authors in clearly presenting a possible new radiologic biomarker to improve pretest probability in making an accurate diagnosis of PCNSL. The serrate sign here is one that ought be easily appreciated by trained neuroradiologists, neurologists, and neurosurgeons. Identifying such imaging patterns that can be easily taught to trainees and incorporated in daily practice with routine brain MRIs. 

The authors mention in line 239-240 that homogenous enhancement is characteristic of PCNSL in immunocompetent patients, however, directly stating that immunocompromised PCNSL are more likely to have ring enhancement may add slightly to the educational aspect of this manuscript. It is not a necessary addition to make but the difference is worthwhile for clinicians to be made aware of. 

Response 1: Thank you for the positive feedback. As you mentioned, considering the educational aspect, we agree that it would be beneficial to highlight the contrasting imaging findings in immunocompromised PCNSL. We have therefore added a section addressing the imaging characteristics of PCNSL in immunocompromised patients.

- Changes in manuscript (dicussion, in line 229-232)

"Homogeneous enhancement on contrast-enhanced imaging is a common characteristic of PCNSL in immunocompetent patients, whereas in immunocompromised patients, PCNSL often presents with a heterogeneous enhancement pattern, frequently featuring ring enhancement [8, 9, 10, 11, 18]."